# Fetal Alcohol Spectrum Disorder and Iron Homeostasis

**DOI:** 10.3390/nu14204223

**Published:** 2022-10-11

**Authors:** Regan Bradley, Koffi L. Lakpa, Michael Burd, Sunil Mehta, Maja Z. Katusic, Jacob R. Greenmyer

**Affiliations:** 1School of Medicine, University of North Dakota, Grand Forks, ND 58201, USA; 2School of Medicine and Public Health, University of Wisconsin, Madison, WI 53706, USA; 3Mayo Clinic, Developmental and Behavioral Pediatrics, Psychiatry and Psychology, Rochester, MN 55905, USA; 4Mayo Clinic, Pediatric and Adolescent Medicine, Rochester, MN 55905, USA

**Keywords:** fetal alcohol spectrum disorders, iron deficiency, iron supplementation, maternal/fetal anemia, maternal/fetal nutrition, pregnancy, prenatal alcohol exposure

## Abstract

Prenatal alcohol exposure results in a spectrum of behavioral, cognitive, and morphological abnormalities collectively referred to as fetal alcohol spectrum disorder (FASD). FASD presents with significant phenotypic variability and may be modified by gestational variables such as maternal nutritional status. Iron serves a critical function in the development of and processes within central nervous system (CNS) structures. Gestational iron deficiency alters CNS development and may contribute to neurodevelopmental impairment in FASD. This review explores the relationship between iron deficiency and fetal alcohol spectrum disorder as described in small animal and human studies. Consideration is given to the pathophysiologic mechanisms linking iron homeostasis and prenatal alcohol exposure. Existing data suggest that iron deficiency contributes to the severity of FASD and provide a mechanistic explanation linking these two conditions.

## 1. Introduction

### 1.1. Prenatal Alcohol Exposure, Maternal Nutrition, and Iron Homeostasis

Alcohol readily distributes across the placenta and enters fetal circulation [1]. Consumption of the teratogen alcohol during the prenatal period can lead to a phenotypically diverse disorder named fetal alcohol spectrum disorders (FASD). FASD is the most common preventable cause of neurodevelopmental delay and affects at least 1% of children in the United States [2]. The most severe form of FASD is fetal alcohol syndrome (FAS), which is characterized by dysmorphic facial features, growth restriction, and restricted brain growth that results in lifelong neurobehavioral impairment [3]. No amount of alcohol has been shown to be safe during pregnancy [4]. The impact of prenatal alcohol exposure (PAE) varies and the multifaceted factors contributing to the severity of the disease are a source of ongoing investigation [5]. Some factors associated with the clinical severity of FASD include socioeconomic status, maternal/fetal alcohol metabolism, maternal age, and maternal weight [6,7].

In addition to its teratogenic effects, gestational alcohol consumption is associated with adverse fetal nutritional status. A growing body of evidence implicates maternal nutrition as a modifying variable in FASD clinical severity. Alcohol can lead to deficiencies in nutrients through caloric replacement, absorptive and metabolic interference [8,9]. Furthermore, PAE impairs placental nutrient transport and blood flow [10]. Animal models have identified an interaction between PAE and multiple nutrients: choline, copper, essential fatty acids, folate, iron, methionine, selenium, and zinc [11,12]. Pre- and post-natal micronutrient supplementation has been associated with improved aspects of memory and cognition [13,14].

The list of nutrients that alcohol interferes with includes the essential mineral iron [15]. Iron plays a critical role in oxygen transport, organ formation, and brain development [16]. Fetal demand regulates the mobilization of iron from maternal circulation [17,18]. This demand increases during later periods of gestation and results in a higher concentration of iron in the fetus compared to the mother [17]. The maternal iron regulatory hormone hepcidin is suppressed during pregnancy [19]. In cases of iron deficiency in pregnant rats, an upregulation of placental iron transport proteins can minimize the severity of fetal anemia [20].

The recommended daily allowance of iron for pregnancy is 27 mg/day (1240 mg total) [21]. Expansion of maternal blood volume and a rapidly growing fetus, both of which occur early in pregnancy, require high levels of iron and make a pregnant woman vulnerable to iron deficiency. Iron deficiency (ID) leads to iron-deficient anemia (IDA) when the iron stores decline to levels that are inadequate for red blood cell formation. Iron deficiency and iron deficiency anemia are common in pregnant women. Optimization of gestational nutrition, including iron status, is an important goal of public health [12]. Iron supplementation has been proven to reduce the risk of iron deficiency and maternal anemia in pregnancy [22]. 

Iron deficiency and iron deficiency anemia adversely affect fetal neurodevelopment [23]. Iron acquired during the third trimester is critical to the development and growth of the fetus [24]. Perinatal iron deficiency is associated with a higher risk of failing to reach educational milestones, decreased motor development, lower intelligence quotient, difficulties with learning and memory, and decreased cognitive capacity [25,26,27,28,29]. Similarly to ID, PAE impairs cognitive and behavioral development [30]. PAE is hypothesized to alter iron homeostasis and thus can alter the hippocampal synaptic plasticity in animals [30]. 

Both PAE and gestational ID have a relatively high incidence, and it is reasonable to suspect they co-occur in a subset of pregnant women [31,32,33]. FASD and gestational ID result in phenotypically overlapping developmental disorders for the child. Different mechanisms have been proposed for the interaction between alcohol and iron homeostasis, including disruption of fetal iron accumulation, alcohol-induced disruption of placental transport of iron, and alcohol-induced disruption of iron absorption and storage potential of the infant [34]. Alcohol may impact iron homeostasis nutritionally and/or mechanistically. The primary goal of this scoping review is to systematically identify and summarize literature that discusses the link between maternal iron status, iron homeostasis, and FASD. 

### 1.2. Normal Iron Metabolism

Iron plays a role in heme synthesis, oxidation-reduction reactions, DNA synthesis, respiration, and energy production [35,36]. Iron is a potent oxidant and contributes to oxidative damage. Iron is normally protein bound to prevent oxidative damage. 

Iron is found in the body in two forms: reduced state (ferric iron, Fe^3+^) and oxidized state (ferrous iron, Fe^2+^). Ferric iron can undergo hydrolysis to produce insoluble ferric hydroxides [37]. Ferrous iron can undergo Fenton chemistry, which results in reactive oxygen species (ROS) production. High levels of ROS lead to oxidative damage [38]. 

The body receives iron from three sources (1) diet, (2) senescent red blood cells (RBCs), and (3) the liver, which stores most of the bodily iron. Humans mainly derive iron from their diet. Dietary iron is found as either heme or non-heme iron. Consumed iron travels to the small intestine to be absorbed by enterocytes of the distal duodenum and proximal jejunum. These cells contain transporters on their apical (lumen-facing) membranes that enable iron absorption. Heme iron is transported into enterocytes via heme carrier protein transporter 1 (HCP1), while non-heme iron by divalent metal transporter 1 (DMT1) [39]. Non-heme iron is found in the ferric iron form. Enterocytes use their membrane-bound duodenal cytochrome B enzyme to reduce ferric iron to ferrous iron. Following its reduction, iron enters enterocytes through DMT1. 

Intracellular ferrous iron within enterocytes has multiple fates. Cells incorporate intracellular ferrous iron into iron-related proteins such as heme and iron-sulfur clusters. An iron storage protein, ferritin (FTN), stores intracellular iron, while iron exporter, ferroportin 1 (FPN1), exports iron into the plasma [40,41]. When iron is exported out of enterocytes, it must be oxidized back to ferric iron in order to be transported. The basolateral membrane of enterocytes contains a ferroxidase protein, hephaestin, which catalyzes irons oxidation. Then, ferric iron is bound by iron transport protein, transferrin (TF), which contains two ferric iron binding sites (diferric-TF) and delivers the iron to target tissues [42].

When diferric-TF reaches its target tissues, it binds to its receptor transferrin receptor 1 (TfR1). This binding forms a diferric-TF-TfR1 complex, which undergoes clathrin-mediated endocytosis [43,44]. Due to the acidic luminal pH of endolysosomes, the intraluminal ferric iron dissociates from the complex. Then, ferric iron is reduced by endolysosomal membrane-bound metalloreductase six-transmembrane epithelial antigen of prostate 3 [45]. Afterward, endolysosomes release ferrous iron into the cytosol via endolysosomal DMT1 [46]. In the cytosol, ferrous iron has three potential fates similar to the ones previously mentioned: (1) storage in FTN, (2) incorporation into heme or ISC, or (3) export via FPN. After the endolysosomal release of iron, TF and TfR1 are recycled back to the plasma membrane [43,44]. TF is released into the extracellular space, and TfR1 awaits at the membrane for the next diferric-TF molecule.

Iron homeostasis is tightly regulated. The human body contains systemic (posttranslational) and cellular (posttranscriptional) methods for maintaining iron homeostasis. Hepcidin is a hepatic peptide that systemically regulates iron [39,40]. It inhibits iron excretion by binding to FPN, promoting FPN internalization and endolysosomal degradation [40,41]. When iron plasma levels are high, the liver synthesizes and releases hepcidin into the plasma [47]. When iron levels are low, hepcidin production is suppressed, and FPN is present to export iron [40]. 

The iron regulatory protein/iron response element (IRP/IRE) system regulates cellular iron levels [39,48]. IRP1/2 are cytosolic RNA-binding proteins that act as cellular sensors of iron. In low iron states, an IRP binds to a 3′ IRE to stabilize mRNA and promote encoding for TfR1 [39]. In high iron states, an IRP binds with a 5′ IRE to prevent the translation of TfR1 and DMT [39,48].

### 1.3. Placental Iron Transportation and Maternal-Fetal Iron Regulation

Iron is required for proper fetal development. Fetal demands for iron change throughout pregnancy [49]. There is a low demand for iron during the first trimester, while in the third trimester, the demand is high [1]. The proposed hierarchy of iron needs during pregnancy is fetus > maternal hematocrit > maternal iron stores [1,49].

The placenta serves as a transport passage for iron delivery from mother to fetus (Figure 1). 

Maternal-placental-fetal iron transport is similar to the intestinal-plasmal iron transport mechanism. Placental trophoblasts contain TfR1 on the apical surface [17]. Diferric-TF in maternal blood binds TfR1 and is internalized into trophoblast [17]. It is unknown how diferric-TF is transported through the placental cells [19,40]. It is possible that cytosolic iron in the syncytiotrophoblasts is chaperoned and delivered to ferroportin for export out of the cell [19]. Alternatively, cytosolic iron in the syncytiotrophoblasts may be delivered to ferritin and released through ferritinophagy [19]. However, ferrous iron is moved through the syncytiotrophoblasts, and it is released from the trophoblast via FPN1 into fetal circulation [1]. Then, ferrous iron is oxidized via ferroxidase zyklopen and bound by fetal TF, where it can be delivered to target tissues such as the liver and RBCs [1]. The placenta regulates iron transport to protect the fetus in the case of maternal iron deficiency; however, alcohol may alter the placental function and possibly impair iron transport.

### 1.4. Blood–Brain Barrier Development and Iron Transportation

The blood–brain barrier (BBB) is a vascular structure that separates the peripheral circulation from the central nervous system and regulates the passage of oxygen, nutrients, molecules, and ions into the brain. The multicellular BBB is composed of pericytes, astrocytes, and brain microvascular endothelial cells (BMVEC). Development of the BBB occurs through three main processes: (1) angiogenesis and barrier induction induced by VEGF and Wnt signaling, (2) sealing of the barrier through interactions of CNS endothelial cells with parenchymal cells, and (3) maturation and maintenance by pericytes and astrocytes [50]. The BBB is formed and functional by the third trimester in humans and late gestation in rodents [51].

BMVEC plays a critical role in transporting iron into the brain. BMVEC express high amounts of TF receptors and accumulate iron via two mechanisms: (1) the canonical TF-iron and TfR mediated pathway and (2) non-holo-TF (NTBI) apical uptake [52]. The canonical TF-iron mediated pathway occurs via clathrin-dependent endocytosis of the TF-TFR complex [52]. Once endocytosed, iron is reduced and released from transferrin in the endosome. Iron exits the endosome and enters the cytoplasm via DMT1. Alternatively, BMVEC can also traffic iron via NTBI uptake at the apical service [52]. In this pathway, NTBI near the apical surface is reduced to Fe^2+^ by ferrireductase. Fe^2+^ enters the cell through DMT1 into the cytoplasm. Although the mechanism is incompletely understood, cytosolic Fe^2+^ may be transferred to the chaperone molecule poly9rC)-binding protein 2 (PCBP2). Cytosolic iron is transferred to FPN for export from BMVEC into the circulation via apical FPN receptors or into the brain via basal FPN receptors. The importance of BMVEC FPN has been demonstrated by (1) lethal outcomes in FPN knockout mice and (2) forebrain and neural tube defects in FPM mutant (*flatiron* (*ffe/ffe*) mice) [41,53]. Reduction of CNS ferroportin expression or function by exposure to alcohol could greatly impair neurodevelopment. 

### 1.5. Animal Models of FASD

The study of FASD in humans is limited by variables that confound gestational alcohol consumption: volume of ethanol exposure, timing of ethanol exposure, socioeconomic environment, maternal health, and maternal diet. Animal models can control for these variables and isolate the effects of alcohol on fetal development. *Caenorhabditis elegans* and zebrafish are common non-mammal models of FASD [54,55]. Nematode and zebrafish models have the advantage of embryo transparency, which allows for precise timing of alcohol delivery and easy observation of physical deformities. Mammal models that utilize mice, rats, sheep, and primates, offer advantages for the study of complex behavior and brain structures. The following variables should be considered with any mammal model of FASD: (1) alcohol exposure patterns, (2) blood alcohol concentration, (3) control group, and (4) route of administration. Rodents are commonly favored for FASD research due to their short gestation, large numbers of offspring, and ease of handling [56]. One of the primary limitations of rodent-based FASD research is that the third-trimester correlation of human development occurs after birth in rodents [56]. There is no third-trimester equivalent in the mouse. In humans, the fetal demand for iron is highest during this period [1,24]. Despite this deficit, rodents are still favored in research. Mice (e.g., C57BL/6 strain) are frequently used because of the availability of existing models, short lifespan, and genetics that resemble humans [56]. Rats are favored by some groups, given their more sophisticated behavior and larger anatomy than mice. Rodent models have contributed greatly to our understanding of the teratogenic effects of alcohol, including compelling data that maternal binge drinking is particularly detrimental to fetal development [56]. In summary, rodent models of FASD offer the ability to control for variables such as maternal nutrition when evaluating neurodevelopmental outcomes of PAE and have greatly contributed to our understanding of FASD. 

## 2. Search Methodology

A search was executed in the following databases: Ovid Cochrane Central Register of Controlled Trials (1991+), Ovid Embase (1974+), Ovid Medline (1946+ including epub ahead of print, in-process and other non-indexed citations), Scopus (1788+), and Web of Science Core Collection (Science Citation Index Expanded 1975+ & Emerging Sources Citation Index 2015+). The following final search terms were included: “fetal alcohol” “fetal alcohol syndrome” “fetal alcohol spectrum disorder” “prenatal alcohol exposure” AND “iron” “iron deficiency”. The search was conducted on 29 March 2022, and organized via the Covidence.org software. Two authors independently reviewed and reached a consensus upon which articles to include. For our formal search methodology study inclusion criteria, studies needed to discuss (1) iron and (2a) prenatal alcohol exposure or (2b) fetal alcohol spectrum disorder. We performed searches of all included literature in included articles to identify additional articles not included in database search results. Studies that did not meet all inclusion criteria, were inaccessible, or were in non-English languages were excluded. The authors used discretion to determine the supporting information to include that was not identified in the scoping search.

## 3. Search Results

Search results are depicted in Figure 2. A total of 3394 studies were originally screened. Fifty-nine of those studies were assessed for eligibility in full-text review. Twenty-seven studies were included in the scoping review, and seven were included in the citation review. Articles discussing both FASD and iron are categorized in Table 1. 

## 4. Literature Synthesis

### 4.1. Animal Studies of Alcohol and Iron in Pregnancy

#### 4.1.1. Neurodevelopment

The rat model of prenatal alcohol exposure provides evidence that maternal iron levels affect the phenotype of offspring exposed to alcohol in utero. Rufer and colleagues (2012) compared the offspring of pregnant, alcohol-consuming rats with an iron deficient (20 ppm) versus iron sufficient (IS) (100 ppm) diet [31]. PAE and maternal ID had a synergistically deleterious effect on associative learning tasks, including auditory-cued conditioning (amygdala), contextual fear conditioning (amygdala and hippocampus), delay eyeblink conditioning (cerebellum), and trace eyeblink conditioning (cerebellum and hippocampus) [31]. Huebner et al. (2015) performed a histologic analysis of offspring rat brains and found reduced cellularity in the cerebellar and hippocampal regions that correlated with gestational iron status and alcohol exposure [32]. The combination of alcohol exposure and ID reduced cerebellar myelin content and increased neuronal apoptosis [32]. Taken together, studies performed by Rufer and Hueber have found that learning deficiencies and neurohistologic abnormalities among PAE rats are greater in ID offspring and lesser in IS offspring [31,32].

The mechanistic interaction between iron and alcohol that alters fetal neurodevelopment is an area of ongoing investigation. White matter has high levels of iron [83]. Glial cells and oligodendrocytes store iron and regulate iron delivery to neurons [83]. Normally high rates of brain iron uptake during early postnatal ages can be diminished when systemic iron availability is lowered [81]. Alcohol may interact with reduced iron status to impair white matter and subsequent neurodevelopment. PAE has multiple effects on the white matter system, including decreased oligodendrocyte differentiation, increased cell death, delayed myelination, reduced white matter content, and disorganization of white matter tracts [84]. 

Miller and colleagues (1995) reported evidence suggesting that PAE alters fetal iron metabolism and impedes brain iron distribution [65]. Their group fed well-nourished pregnant rats high dose alcohol during gestational days 11–20 [65]. PAE altered the normal developmental patterns of ferritin and transferrin and was associated with a slowed response rate of these iron regulatory proteins [65]. Adolescent and adult rats who had prenatal alcohol exposure had (a) reduced brain iron concentrations and (b) reduced cerebral cortex iron mobility [65]. Thus, the aberrant effects of PAE on iron homeostasis in the brain were sustained long after the neonatal period. Later studies corroborated the effect of PAE on brain iron levels. Huebner (2016) fed pregnant rats an ID (4–5 ppm) or IS (100 ppm) diet and ethanol (5.0 g/kg) or isocaloric maltodextrin [61]. ID decreased iron and ferritin content of the fetal brain, fetal liver, and maternal liver by nearly 200% [61]. In both ID and IS groups, PAE decreased brain iron (15–20%) [61].

Rat-based studies of neurodevelopment demonstrate that the combination of PAE and ID (1) reduces brain iron concentration, cellularity, and myelin content; (2) alters iron-binding protein response; and (3) impairs associative learning tasks.

#### 4.1.2. Iron Regulating Genes, Storage, and Indices

Hepcidin is a possible link between alcohol consumption and iron overload in adults. In non-pregnant adults, chronic alcohol use suppresses liver hepcidin expression and limits ferroportin degradation [5]. Mechanistically, this could be driven by (1) an inflammatory response with cytokines such as IL-6 or (2) liver hypoxia [5]. Chronic alcohol abusers have higher serum levels of free iron than those with former alcohol abuse status [5]. Excess circulating and stored iron caused by alcohol consumption may be an important step in alcohol-induced liver damage. Alcohol consumption in non-pregnant adults leads to suppression of hepcidin expression; conversely, alcohol consumption during pregnancy stimulates expression of hepcidin. We do not know how pregnancy influences hepcidin expression in adult females consuming alcohol. 

Fetal liver hepcidin is the primary regulator of fetal iron status [82]. In Huebner’s 2016 rat studies, PAE greatly increased hepatic hepcidin mRNA expression in both the mother and fetus (>300%) and increased fetal liver iron by 30–60% [61]. Pregnant rats fed ID or IS diets had normal hematologic values, but, independent of iron status, alcohol-exposed fetuses had reduced hemoglobin, hematocrit, and red blood cells [61]. PAE offspring had inadequate erythropoiesis even though PAE increases the iron content of the liver and fetal erythropoiesis is mostly hepatic [61]. This concept is supported by another study that found alterations to the erythropoietic system [68]. In iron deficient rats, PAE was also associated with reduced brain content of ferritin, transferrin, and transferrin receptor compared to maltodextrin. Taken together, these results in rats suggest that PAE alters erythropoiesis and may disrupt normal iron binding protein response. One proposed mechanism for increased hepcidin production in rats with PAE is the proinflammatory effects of alcohol [5]. Alcohol stimulates cytokine production in chronic users and binge drinkers [85]. Anemia of chronic inflammation is driven by proinflammatory cytokines such as IL-6 that act through the JAK/STAT pathway to increase hepcidin transcription [5,67]. Gestational alcohol may act as a proinflammatory molecule, sustain an elevation in hepcidin, sequester iron in the liver, reduce placental iron transport, and impair erythropoiesis. 

A sheep model of PAE provided partially discrepant data to the rat studies described above [69]. Daily ethanol exposure in sheep decreased fetal mRNA levels of hepcidin and fetal liver ferric iron content [69]. Placental ferritin mRNA levels were not altered by ethanol exposure [69]. Ethanol exposure reduced levels of circulating iron in fetal circulation [69]. These data suggest that iron may have been sequestered in the maternal liver. Alcohol may induce hepcidin suppression through stabilization of hepatocyte hypoxia-inducible factor (HIF) [5,86]. It is difficult to reconcile the rat and sheep data. While both models provide evidence for PAE’s ability to alter iron homeostasis, hepcidin mRNA and fetal liver content were increased in rats and decreased in sheep. There is any number of variables that could contribute to the variation of results, including the sheep model’s moderate alcohol exposure (compared to high levels in rats) and herbivore metabolism [5]. 

Studies of the amount of iron in the tissues of pregnant rats exposed to alcohol have reported varying results. Two studies reported no changes in iron levels in samples such as fetal serum, maternal endometrium, maternal liver, and maternal femur [57,59]. By contrast, other studies have shown changes in iron levels in tissues, including higher maternal levels of iron absorption and liver iron, and higher neonatal liver and femur iron [32,58,64].

#### 4.1.3. Weight and Growth

Rats with PAE present with significantly reduced fetal total, brain, liver, and placental weight compared with those without PAE [61,62,63,65]. In one study, rats with ID-PAE had reduced body and liver weights compared to those fetuses with IS-PAE [61]. Another study showed a significant reduction in liver iron concentration in rats consuming both alcohol and iron compared to those just being fed iron without alcohol [66]. In another study, in both male and female offspring, maternal ID reduced postnatal growth; in male offspring, PAE interacted with maternal ID to further restrict growth [32]. PAE significantly reduced placental weight and placental efficiency; the lowest placental efficiency was found in ID-PAE dams [63]. The placenta efficiency was higher (but not statistically significant) in the iron-fortified PAE dams [63]. The placental-to-fetal body weight ratio was significantly higher in PAE pregnancy, and iron fortification did not significantly mitigate this. These rat-based studies have demonstrated that the combination of PAE and ID reduces fetal total weight, liver weight, brain weight, placental size, and placental efficiency. 

#### 4.1.4. Supplementation

Iron distribution is physiologically prioritized to the red blood cells and the developing brain of the fetus over storage in the liver. This ratio is significantly skewed in PAE pregnancy, so the liver storage of iron is greater than red blood cells and brain iron [62]. Iron fortification in PAE pregnancy helps correct iron distribution in the fetus toward a physiologic distribution [62]. 

Low iron reserves in the mother can lead to significant reductions in body growth, learning, myelination, and survival of neurons in the fetus [31]. However, when these iron stores in the mother are adequate, the neurodevelopmental damage caused by PAE is mitigated [31]. Supplementation of iron can overcome ID caused by PAE and reverse fetal anemia, increase brain iron, increase brain weight, and normalize hepcidin expression in rat models [60,62].

### 4.2. Iron and Alcohol in Pregnancy: Population Studies

#### 4.2.1. Extent of the Problem

Numerous studies performed in the United States, Europe, Africa, and Asia have described an iron-deficient diet in populations with a high prevalence of gestational alcohol use [8,34,72,73,74,78,80,82]. May and colleagues have published multiple papers that reported inadequate dietary intake of mothers of children with FASD [74,75]. Dietary intake is deficient in micronutrients such as iron in greater than 50% of mothers of children with FASD [74]. In a cross-sectional analysis, only 9.1% of participants with gestational alcohol use were taking iron supplementation at the time of study enrollment [78]. A minority of women consuming alcohol take supplements before and early in pregnancy. More than 85% of pregnant women with opioid and/or alcohol use during pregnancy reported inadequate intake of iron among ten other nutrients (fiber, calcium, copper, iodine, zinc, choline, folate, vitamin C, and vitamin D) [72]. Heavy drinkers may be at a higher risk for micronutrient deficiencies due to problems with absorption and utilization [72]. Most women (87.2%, n = 107/123) received prenatal iron supplementation late in pregnancy [72]. 

Brooten and colleagues (1987) conducted a study in which 13% (n = 42/326) of pregnant women reported regular alcohol consumption, and 2% (n = 7/326) of women reported consuming more than two drinks daily [70]. Of this sample, dietary iron intakes ranged from 54–62% of the RDA [70]. In a study of 1398 pregnant women, low iron status (measured by ferritin) was associated with the heaviest drinking category of ≥8 drinks per day (0.6%, n = 9/1398) [80]. In the subset with the heaviest drinking, there was a significantly higher proportion of iron depletion (ferritin less than 12 ng/mL or transferrin saturation less than 16%) compared to the women at lower alcohol use levels [80]. Only 11% of the heaviest drinkers were anemic (hemoglobin concentration of 11 g/100 mL or less), which was similar to the other study groups [80]. After accounting for relevant characteristics such as the mother’s age, education, and nicotine use, there was no significant association between alcohol intake and ferritin levels. Maternal nutritional status in pregnant alcohol users may be influenced by demographic and socioeconomic factors. 

These data show that (1) iron deficiency and gestational alcohol use co-occur, (2) maternal diets are often deficient in micronutrients, (3) heavy drinkers are at greater risk for iron deficiency, and (4) demographic and socioeconomic variables impact the nutritional status of pregnant alcohol users.

#### 4.2.2. Neurodevelopment

Commensurate with rodent model findings, humans with prenatal exposure to alcohol have altered brain iron distribution. Nakhid and colleagues (2022) used T1-weighted and quantitative susceptibility mapping (QSM) MRI scans to indirectly measure brain iron in children aged 7.5–15 years old [77]. Brains exposed to alcohol prenatally have lower iron susceptibility in the hippocampus [77]. However, this finding did not survive corrections done in the study. The researchers stated they did not find the hippocampus susceptibility to being related to mental health symptoms but may be associated with cognitive difficulties commonly present in individuals with PAE [77]. These changes to the brain accounted for internalizing (thalamus) and externalizing (putamen) symptoms [77]. 

Maternal iron status is associated with neurobehavioral and cognitive outcomes in patients with FASD. Maternal iron deficiency strongly correlated with worsened outcomes in FASD in a clinical cohort of 96 infants with prenatal alcohol exposure [77]. Another study found a positive association between maternal iron intake and the child’s IQ at seven years of age [75]. Human and animal studies demonstrate that iron homeostasis and iron status play a role in neurodevelopmental outcomes of FASD.

Neurodevelopmental damage by alcohol is mitigated by adequate iron stores [31]. Alcohol and ID may reduce myelination in brain regions correlated with executive function, math processing, visual perception, visual–motor integration, and learning. Other studies have also shown that PAE and ID synergize to impair delayed eyeblink conditioning and impair contextual fear conditioning [31,32].

#### 4.2.3. Iron Deficiency and Iron Deficiency Anemia

Carter and Colleagues, who first reported the association between infant DIA with PAE, found that infants whose mothers binge drank during pregnancy (>4 drinks per occasion) were 3.6 times more likely to be diagnosed with IDA at twelve months of age than infants whom mothers abstained or drank less [34]. A blinded, prospective, longitudinal study found that PAE was associated with lower hemoglobin (0.3 g/dL lower among infants born to heavy drinkers than controls) and higher risk of ID and IDA at 6.5 months [73]. In this same sample, children with FAS or PFAS diagnosis had higher prevalences of anemia and IDA than nonsyndromal PAE-exposed and control children [73]. The study also detected a shift of iron into storage over erythropoiesis or placental transfer due to lower hemoglobin:log(ferritin), higher hepcidin, and increased prevalence of anemia of inflammation, all associated with drinking frequency [73]. This sequestration into storage manifested in higher neonatal ferritin levels and lower neonatal hemoglobin:log(ferritin) [73]. In a prospective longitudinal birth cohort of 206 pregnant women, maternal dietary iron intake was associated with higher hemoglobin among control mothers and not drinking mothers [73]. As described above, rat models of FASD have demonstrated increased hepatic hepcidin, increased fetal liver iron, and reduced fetal hemoglobin in alcohol-exposed pregnancies [61]. Thus, PAE is associated with lower hemoglobin and a higher prevalence of ID/IDA in both human and rodent models. 

#### 4.2.4. Weight and Growth

FASD is known to cause growth restriction. Carter and colleagues (2012) showed that heavy PAE could restrict growth and cause leaner body compositions [71]. Iron deficiency is found at higher rates in individuals with FASD who are growth restricted. In one sample, children with FASD and the slowest growth trajectories had the highest prevalence of IDA [34]. Carter and colleagues (2022) showed that mothers with poorer dietary iron intakes had more severe fetal and postnatal growth restrictions compared to those mothers with better iron intake [87]. The growth restrictions were only evident in the most poorly nourished mothers. This reinforces the importance of nutritional evaluation in children with FASD—especially those with growth restrictions. 

#### 4.2.5. Supplementation

The effects of ID appear to be more severe and less reversible when ID occurs in the first half of the pregnancy [82]. Iron supplementation in pre-pregnancy, times of trying to conceive, and immediately upon the knowledge of pregnancy can mitigate some neurodevelopmental effects of ID [82]. Strong consideration should be given to iron supplementation in pregnant women with gestational alcohol use. In Carter’s 2021 study, the vast majority of mothers reported good adherence to antenatally prescribed iron supplementation (96.2% adherence in heavy drinkers) [73].

Despite lower hemoglobin values in patients with FASD, iron supplementation is extremely uncommon in this population. Nineteen children assessed at an FASD Diagnostic Clinic were questioned to find that 92.9% of the study participants were adequate in iron intake, but none reported iron supplementation [88]. 

Choline supplementation in children with FASD is associated with improved behavioral symptoms, non-verbal intelligence, verbal memory, and visual-spatial skills [89]. Choline supplementation has been particularly beneficial for the improvement in the behaviors of rats with iron deficiency [90]. The next generation in rats demonstrated that (1) iron deficiency causes dysregulated expression in genes associated with neuropsychiatric conditions and (2) choline supplementation helps correct the dysregulated gene expression caused by ID. These studies suggest that ID and choline supplementation are two promising interventions for increasing the neurodevelopmental outcomes of patients with FASD.

## 5. Conclusions

Evidence for an interaction between PAE and altered iron homeostasis is derived from both animal and human studies. Rat-based studies have demonstrated that the combination of PAE and ID (1) reduces brain iron concentration, cellularity, and myelin content; (2) alters iron-binding protein response; (3) impairs associative learning tasks; (4) alters erythropoiesis; (5) disrupts normal iron binding protein response; (6) reduces fetal total weight, liver weight, brain weight, and placental size; and (7) impairs placental efficiency. Iron supplementation in rat models of PAE can (1) overcome ID; (2) reverse fetal anemia; (3) increase brain iron; (4) increase brain weight; (5) normalize hepcidin expression; (6) improve iron distribution; and (7) reduce neurodevelopmental impairment. 

Human studies of PAE have found (1) a relationship between gestational alcohol use and insufficient iron intake; (2) increased frequency of ID in infants with high levels of PAE; (3) slower growth in infants with ID-PAE compared to IS-PAE; and (4) a correlation between IQ of infants born to mothers with gestational alcohol and levels of iron in the gestational diet. Iron and choline supplementation may help mitigate neurodevelopmental defects of PAE. 

PAE and gestational iron deficiency are independently significant public health issues. Iron deficiency in alcohol-exposed pregnancy has effects that extend beyond gestation into the life of offspring. We suggest that maternal nutritional status, including iron, and PAE, may compound and explain some of the neurodevelopmental phenotypic variances in FASD. Further research is necessary to (1) clarify the mechanisms underlying the interactions between PAE and ID; (2) test whether iron supplementation in humans mitigates adverse effects of PAE in randomized controlled trials; (3) identify optimum levels of iron intake in pregnant women with alcohol consumption to facilitate fetal development; and (4) establish a reliable reference range for iron status biomarkers in pregnant women with and without alcohol consumption. Centers that provide substance use care to pregnant women should consider comprehensive nutritional assessment in their evaluation and management. Maternal iron supplementation may be a way to mitigate the impact of alcohol on the fetus and reduce long-term effects of FASD. Pediatricians caring for infants with concerns about intra-uterine alcohol exposure should consider promoting iron supplementation, encouraging iron-rich diets, and conducting anemia screening at age-appropriate time intervals. 

## Figures and Tables

**Figure 1 nutrients-14-04223-f001:**
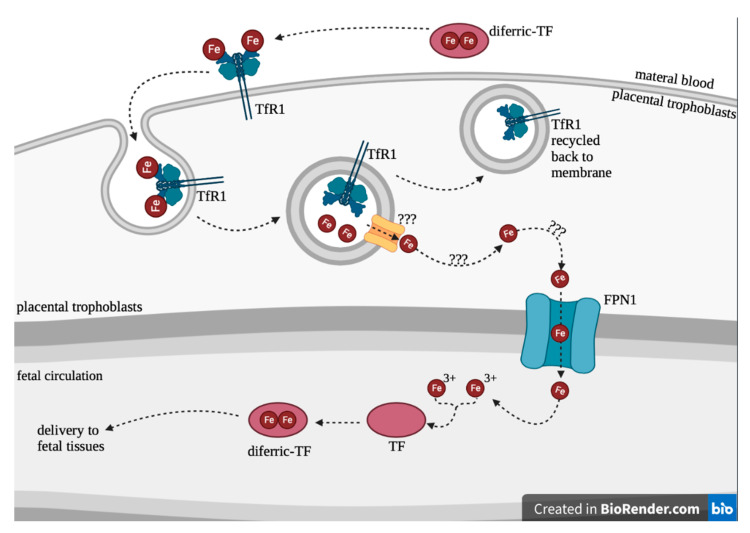
Placental iron transportation.

**Figure 2 nutrients-14-04223-f002:**
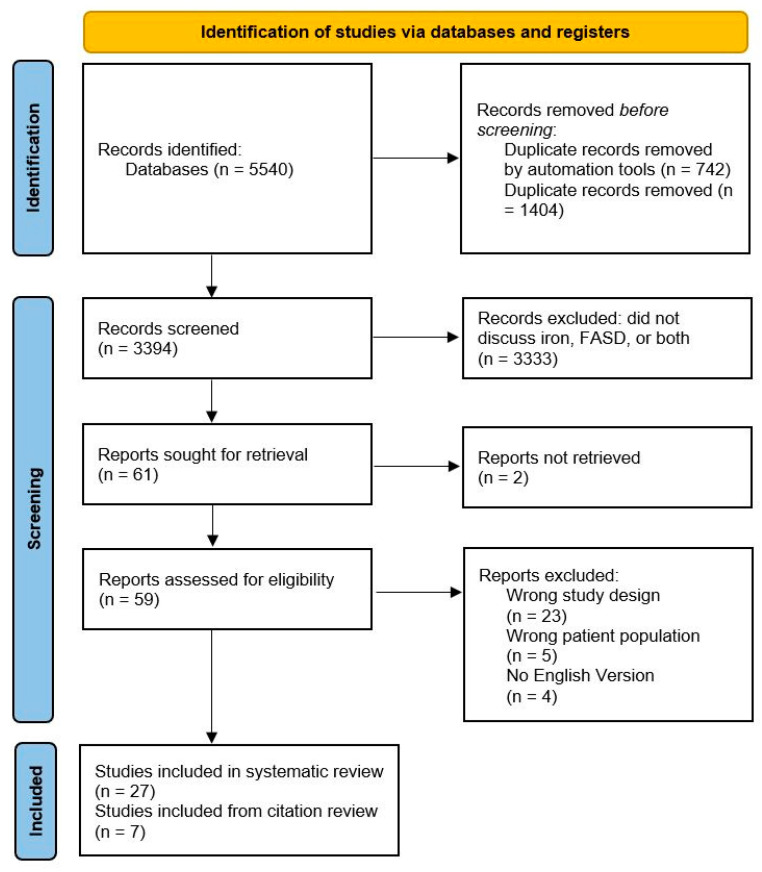
PRISMA flowchart for scoping review.

**Table 1 nutrients-14-04223-t001:** Scoping review results by category.

	Author & Year	Title	Citation
Basic Science/Animal Model Studies	Amini 1995	Maternal hepatic, endometrial, and embryonic levels of Zn, Mg, Cu, and Fe following alcohol consumption during pregnancy in QS mice	[57]
	De La Fuente-Oretega 2019	Prenatal ethanol exposure misregulates genes involved in iron homeostasis promoting a maladaptation of iron dependent hippocampal synaptic transmission and plasticity	[30]
	Gordon 1984	Effects of prenatal ethanol exposure on iron utilization in the rat	[58]
	Heil 1999	Ethanol and lactation: Effects on milk lipids and serum constituents	[59]
	Helfrich 2022	Gestational Iron Supplementation Improves Fetal Outcomes in a Rat Model of Prenatal Alcohol Exposure	[60]
	Huebner 2016	Prenatal Alcohol Exposure Alters Fetal Iron Distribution and Elevates Hepatic Hepcidin in a Rat Model of Fetal Alcohol Spectrum Disorders	[61]
	Huebner 2015	Maternal iron deficiency worsens the associative learning deficits and hippocampal and cerebellar losses in a rat model of fetal alcohol spectrum disorders	[32]
	Huebner 2018	Dietary Iron Fortification Normalizes Fetal Hematology, Hepcidin, and Iron Distribution in a Rat Model of Prenatal Alcohol Exposure	[62]
	Kwan 2020	Maternal iron nutriture modulates placental development in a rat model of fetal alcohol spectrum disorder	[63]
	Mendelson 1980	The effect of duration of alcohol administration on the deposition of trace elements in the fetal rat	[64]
	Miller 1995	Iron regulation in the developing rat brain: effect of in utero ethanol exposure	[65]
	Olynyk 1995	A long-term study of the interaction between iron and alcohol in an animal model of iron overload	[66]
	Rufer 2012	Adequacy of maternal iron status protects against behavioral, neuroanatomical, and growth deficits in fetal alcohol spectrum disorders	[31]
	Saini 2019	Alcohol’s Dysregulation of Maternal–Fetal IL-6 andp-STAT3 Is a Function of Maternal Iron Status	[67]
	Sanchez 1998	Effect of chronic ethanol administration on iron metabolism in the rat	[68]
	Sozo 2013	Alcohol exposure during late ovine gestation alters fetal liver iron homeostasis without apparent dysmorphology	[69]
Population/Human Studies	Brooten 1987	A survey of nutrition, caffeine, cigarette and alcohol intake in early pregnancy in an urban clinic population	[70]
	Carter 2007	Fetal alcohol exposure, iron-deficiency anemia, and infant growth	[34]
	Carter 2012	Effects of Heavy Prenatal Alcohol Exposure and Iron Deficiency Anemia on Child Growth and Body Composition through Age 9 Years	[71]
	Carter 2017	Maternal Alcohol Use and Nutrition During Pregnancy: Diet and Anthropometry	[72]
	Carter 2021	Prenatal alcohol-related alterations in maternal, placental, neonatal, and infant iron homeostasis	[73]
	May 2014	Dietary intake, nutrition, and fetal alcohol spectrum disorders in the Western Cape Province of South Africa	[74]
	May 2016	Maternal nutritional status as a contributing factor for the risk of fetal alcohol spectrum disorders	[75]
	Molteno 2014	Infant Emotional Withdrawal: A Precursor of Affective and Cognitive Disturbance in Fetal Alcohol Spectrum Disorders	[76]
	Nakhid 2022	Brain Iron and Mental Health Symptoms in Youth with and without Prenatal Alcohol Exposure	[77]
	Shrestha 2018	Dietary Intake Among Opioid- and Alcohol-Using Pregnant Women	[78]
	Skalny 2016	The effect of alcohol consumption on maternal and cord blood electrolyte and trace element levels	[79]
	Streissguth 1983	Alcohol use and iron status in pregnant women	[80]
Review Articles	Cogswell 2003	Cigarette smoking, alcohol use and adverse pregnancy outcomes: implications for micronutrient supplementation	[8]
	Connor 1994	Iron acquisition and expression of iron regulatory proteins in the developing brain: Manipulation by ethanol exposure, iron deprivation and cellular dysfunction	[81]
	Helfrich 2018	Maternal iron nutriture as a critical modulator of fetal alcohol spectrum disorder risk in alcohol-exposed pregnancies	[5]
	McArdle 2014	Iron deficiency during pregnancy: the consequences for placental function and fetal outcome	[82]
	Naik 2022	Effects of nutrition and gestational alcohol consumption on fetal growth and development	[15]
	Sebastiani 2018	The effects of alcohol and drugs of abuse on maternal nutritional profile during pregnancy	[9]

## Data Availability

Literature was obtained from publically available databases and registers.

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
