# Peer review of "Fetal Alcohol Spectrum Disorder and Iron Homeostasis"

_nutrients, 2022, doi:10.3390/nu14204223_

Round 1
Reviewer 1 Report
This is a review paper that attempts to harmonize across animal and human studies the impact of prenatal ethanol exposure (or use) on iron levels in brain and body. Furthermore an attempt is made to provide thoughts on how this may alter the physiology, neurobiology and behavior. In general the paper is fine although the depth of the scholarship and the overall application component could be developed more. The initial review of FAS in the animal literature is brief and needs to be developed. the statement that on line 30-32 about first trimester exposure being worse is not really true. Exposure during different times impacts different bodily systems worse. For example, Goodlett using rat models has demonstrated that the hippocampus is differentially effected by when ethanol is exposure...later exposure is worse than earlier. Thomas has some nice data on choline supplements improving animals exposed to ethanol, this data should be included and thought given to how it might alter iron levels. The second half of the paper takes on a more laundry list of issues and the integration to the rest of the paper is lacking (body weight paragraph for example).
Author Response
This is a review paper that attempts to harmonize across animal and human studies the impact of prenatal ethanol exposure (or use) on iron levels in brain and body. Furthermore an attempt is made to provide thoughts on how this may alter the physiology, neurobiology and behavior. In general the paper is fine although the depth of the scholarship and the overall application component could be developed more.
Thank you for your review of our manuscript and your thoughtful comments. We appreciate the reviewer’s assessment that the overall application of this review could be developed more. We have made multiple additions to the original manuscript that we help improve the application of this manuscript and put our discussion of FASD/PAE and iron homeostasis in context. We think that these additions improve the strength of the manuscript significantly.
- Figure -1: Placental iron transport
- Discussion of blood brain barrier (BBB) development and iron across the BBB
- General summary of FAS in animal literature
- Discussion on choline supplementation and its potential mechanistic interaction with iron
The initial review of FAS in the animal literature is brief and needs to be developed.
Thank you for this comment. We have expanded our initial review of FAS in the animal literature. We are pleased with the improvement of our article’s applicability with this new addition.
We have added the following information that provides an overview of animal literature of FASD:
“1.4 Animal models of FASD
The study of FASD in humans is limited by variables that confound gestational alcohol consumption: volume of ethanol exposure, timing of ethanol exposure, socioeconomic environment, maternal health, and maternal diet. Animal models can control for these variables and isolate the effects of alcohol on fetal development. Caenorhabditis elegans and zebrafish are common non-mammal models of FASD [88,89]. Nematode and zebrafish models have the advantage of embryo transparency, which allows for precise timing of alcohol delivery and easy observation of physical deformities. Mammal mod-els that utilize mice, rats, sheep, and primates, offer advantages for the study of complex behavior and brain structures. The following variables should be considered with any mammal model of FASD: (1) alcohol exposure patterns, (2) blood alcohol concentration. (3) control group, and (4) route of administration. Rodents are commonly favored for FASD research due to their short gestation, large numbers of offspring, and ease of handling [90]. One of the primary limitations of rodent based FASD research is that the third trimester correlate of human development occurs after birth in rodents. Mice (e.g., C57BL/6 strain) are frequently used because of the availability of existing models, short lifespan, and genetics that resemble humans [90]. Rats are favored by some groups given their more sophisticated behavior and larger anatomy than mice. Rodent models have contributed greatly to our understanding of the teratogenic effects of alcohol, including compelling data that maternal binge-drinking is particularly detrimental to fetal development [90]. In summary, rodent models of FASD offer the ability to control for variables such as maternal nutrition when evaluating neurodevelopmental outcomes of PAE and have greatly contributed to our understanding of FASD.”
The statement that on line 30-32 about first trimester exposure being worse is not really true. Exposure during different times impacts different bodily systems worse. For example, Goodlett using rat models has demonstrated that the hippocampus is differentially affected by when ethanol is exposure...later exposure is worse than earlier.
Thank you for bringing up this error. We have deleted the statement about first trimester exposure being worse (lines 30-32).
Thomas has some nice data on choline supplements improving animals exposed to ethanol, this data should be included and thought given to how it might alter iron levels.
Thank you for this idea. We agree that a discussion of choline supplementation fits nicely into our review article. We have added choline into our review in several places. The largest section is copied below:
Choline supplementation in children with FASD is associated with improved behavioral symptoms, non-verbal intelligence, verbal memory, and visual-spatial skills [91]. Choline supplementation has been particularly beneficial for improvement in behaviors of rats with iron deficiency [92]. Next generation in rats demonstrated that (1) iron deficiency causes dysregulated expression in genes associated with neuropsychiatric conditions and (2) choline supplementation helps correct the dysregulated gene expression caused by ID. These studies suggest that ID and choline supplementation are two promising interventions for increasing the neurodevelopmental outcomes of patients with FASD.
The second half of the paper takes on a more laundry list of issues and the integration to the rest of the paper is lacking (body weight paragraph for example).
We agree with the reviewer’s assessment that the second half of the paper was not appropriately integrated and did not appropriately organize the issues that it was trying to address. We substantially revised the second half of our manuscript, including the following changes:
- Created additional sections for the section describing human studies
- Re-organized the human study section according to these sections
- Deleted several sentences in the human studies section that seemed out of place or were not well integrated into the paragraph/section
- Re-wrote poorly worded or unclear sentences in the second half of the manuscript
- Referenced animal data in the human data section to better bridge the gap between animal and human data
Reviewer 2 Report
In the manuscript entitled “Fetal alcohol spectrum disorder and iron homeostasis”, the authors discuss current and historical literature on the connection between fetal alcohol spectrum disorder and iron homeostasis. The topic discussed is relevant to Nutrients. The authors do an exemplary job at summarizing the limited literature and making connections to different topics. There are a number of specific critiques that authors could address to improve the paper.
1. The manuscript would benefit from including a figure to outline the transport of iron across the placenta barrier. In combination with the text will aid in the readers visualization of this important and topical pathway.
2. The overall writing could be improved with the inclusion of smoother transitions. There are a number of places where two sentences do not flow very well leaving the reader a bit confused on the purpose. Examples include lines 99, 286, 300, and 332.
3. The timing of the blood-brain barrier formation in fetal development, as well as mention of the blood-brain barrier iron transport, would be of interest.
4. The paragraph regarding IRE/IRP regulation (lines 136-141) is a bit oversimplified in the opinion of this reviewer. It may be beneficial to state that IRP binding to a 3’ IRE stabilizes the transcript for translation while binding to 5’ prevents translation.
5. Lines 158-162 are a bit confusing and should be clarified.
Author Response
In the manuscript entitled “Fetal alcohol spectrum disorder and iron homeostasis”, the authors discuss current and historical literature on the connection between fetal alcohol spectrum disorder and iron homeostasis. The topic discussed is relevant to Nutrients. The authors do an exemplary job at summarizing the limited literature and making connections to different topics. There are a number of specific critiques that authors could address to improve the paper.
Thank you for this positive feedback. We are hopeful that our review is relevant, timely, and can contribute to the literature in a significant way. We want to thank the reviewer the specific and thoughtful feedback that we feel helped greatly improve the strength of the manuscript. A point-by-point response is listed below.
- The manuscript would benefit from including a figure to outline the transport of iron across the placenta barrier. In combination with the text will aid in the readers visualization of this important and topical pathway.
Thank you for this suggestion. We agree that the organization and comprehensiveness of our original manuscript would benefit greatly from a figure outlining transport of iron across the placenta. We have added a new figure to our manuscript and are pleased with this addition.
- The overall writing could be improved with the inclusion of smoother transitions. There are a number of places where two sentences do not flow very well leaving the reader a bit confused on the purpose. Examples include lines 99, 286, 300, and 332.
We thank the reviewer for this detailed feedback. We agree that there were several places where transitions needed to be improved. There seemed to be issues with formatting that led to this issue in several places of the manuscript and several places that writing itself could have been improved. We addressed the specific concerns of the reviewer in the following manner:
Line 99: Deleted
Line 286: We have rewritten this portion of the manuscript. This paragraph is now the following:
“Iron distribution is physiologically prioritized to the red blood cells and the develop-ing brain of the fetus over storage in the liver. This ratio is significantly skewed in PAE pregnancy so that the liver storage of iron is greater than red blood cell and brain iron[65]. Iron fortification in PAE pregnancy helps correct iron distribution in the fetus toward a physiologic distribution [65]”.
Line 300: Deleted two preceding sentences and combined line 286 with the proceeding paragraph. This section now is written as:
“Numerous studies have described an iron deficient diet in populations with a high prevalence of gestational alcohol use [9,32,55,71–75]. Most studies on this topic were per-formed in the United States, Europe, Africa, or Asia. May and colleagues have published multiple papers that reported inadequate dietary intake of mothers of children with FASD. In one study of 43 mothers of children with FASD, dietary intake in greater than 50% of these mothers were inadequate in numerous micronutrients including iron[75]. In another study, they asked 57 mothers of children diagnosed with FASD to recall their dietary in-take during pregnancy and found that many of the women were deficiency in numerous nutrients including iron[76].
Line 332: We rearranged the section entitled 4.2.1 Extent of the Problem to have more cohesiveness. We added the following summary statement following these two paragraphs:
“These data show that (1) iron deficiency and gestational alcohol use co-occur, (2) ma-ternal diets often lack adequate micronutrients, (3) heavy drinkers are at greater risk for iron deficiency, and 4) demographic and socioeconomic variables impacts nutritional sta-tus of pregnant alcohol users.”
In addition to addressing the 4 lines noted by the reviewer, we reviewed all transitions in the manuscript to identify areas where transitions or summaries could be improved. This review led to numerous edits (>10) being made throughout the manuscript. We feel that these changes have greatly improved the quality of our writing and the readability of our manuscript. We also improved the organization of our section on human studies with more detailed section headings.
- The timing of the blood-brain barrier formation in fetal development, as well as mention of the blood-brain barrier iron transport, would be of interest.
Both reviewers indicated that expansion of the introduction for proper contextualization of the scoping review would benefit this manuscript. We agree. Part of our expansion of the introduction includes a section on development of the BBB and the role of iron transport in the BBB.
We added the following section on the BBB to our manuscript:
“The blood-brain barrier (BBB) is a vascular structure that separates the peripheral circulation from the central nervous system and regulates the passage of oxygen, nu-trients, molecules, and ions into the brain. The multicellular BBB is composed of peri-cytes, astrocytes, and brain microvascular endothelial cells (BMVEC). Development of the BBB occurs through three main processes: (1) angiogenesis and barrier induction induced by VEGF and Wnt signaling, (2) sealing of the barrier through interactions of CNS endothelial cells with parenchymal cells, and (3) maturation and maintenance by pericytes and astrocytes [83]. The BBB is formed and functional by the third trimester in humans and late gestation in rodents [84].
BMVEC play a critical role in transporting iron into the brain. BMVEC express high amounts of TF receptors and accumulate iron via two mechanisms: (1) the canoni-cal TF-iron and TfR mediated pathway and (2) non-holo-TF (NTBI) apical uptake [85]. The canonical TF-iron mediated pathway occurs via clathrin-dependent endocytosis of the TF-TFR complex [85]. Once endocytosed, iron is reduced and released from trans-ferrin in the endosome. Iron exits the endosome and enters the cytoplasm via DMT1. Alternatively, BMVEC can also traffic iron via NTBI uptake at the apical service [85]. In this pathway, NTBI near the apical surface is reduced to Fe2+ by ferrireductase. Fe2+ enters the cell through DMT1 into the cytoplasm. Although the mechanism is incom-pletely understood, cytosolic Fe2+ may be transferred to the chaperone molecule poly9rC)-binding protein 2 (PCBP2). Cytosolic iron is transferred to FPN for export from BMVEC into the circulation via apical FPN receptors or into the brain via basal FPN receptors. The importance of BMVEC FPN has been demonstrated by (1) lethal outcomes in FPN knockout mice and (2) forebrain and neural tube defects in FPM mu-tant (flatiron (ffe/ffe) mice [86,87]. Reduction of CNS ferroportin expression or function by exposure to alcohol could greatly impair neurodevelopment.”
- The paragraph regarding IRE/IRP regulation (lines 136-141) is a bit oversimplified in the opinion of this reviewer. It may be beneficial to state that IRP binding to a 3’ IRE stabilizes the transcript for translation while binding to 5’ prevents translation.
Thank you for helping us address our oversimplification of this process. We have made the following change:
“In low iron states, an IRP binds to a 3’ IRE to stabilize mRNA and promote encoding for TfR1 [37]. In high iron states, an IRP binds with a 5’ IRE to prevent translation of TfR1 and DMT[37,46].”
- Lines 158-162 are a bit confusing and should be clarified.
We agree with the reviewer that lines 158-162 were unclear. We modified this portion of the manuscript. We removed several preceding statements that added little value. We modified lines 158-162 to the succinct summary:
“The placenta regulates iron transport to protect the fetus in the case of maternal iron deficiency; however, alcohol may alter the placental function and possibly impair iron transport.”
Round 2
Reviewer 1 Report
nil
Author Response
Thank you for your positive comments.